# GOING BEYOND FAMILIAR FEATURES FOR DEEP ANOMALY DETECTION

## ABSTRACT

Anomaly Detection (AD) is a critical task that involves identifying observations that do not conform to a learned model of normality. Prior work in deep AD is predominantly based on a *familiarity hypothesis*, where *familiar* features serve as the reference in a pre-trained embedding space. While this strategy has proven highly successful, it turns out that it causes consistent false negatives when anomalies consist of truly *novel* features that are not well captured by the pre-trained encoding. We propose a novel approach to AD using explainability to capture novel features as unexplained observations in the input space. We achieve strong performance across a wide range of anomaly benchmarks by combining similarity and novelty in a hybrid approach. Our approach establishes a new state-of-the-art across multiple benchmarks, handling diverse anomaly types while eliminating the need for expensive background models and dense matching. In particular, we show that by taking account of novel features, we reduce false negative anomalies by up to 40% on challenging benchmarks compared to the state-of-the-art. Our method gives visually inspectable explanations for pixel-level anomalies.

## 1 INTRODUCTION

Anomaly detection (AD) is a crucial task that involves identifying abnormal samples in test data by learning patterns from normal training samples. In real-world applications, the occurrence of anomalies is often unpredictable and can lead to severe consequences, to the extent that AD has been identified as a critical component in improving organizational security under Catastrophic AI Risks. Hendrycks et al. (2023). Anomaly detection is used across diverse domains, including quality control in manufacturing, medical imaging for early disease diagnosis, enhancing security and surveillance systems, and improving the robustness of AI models. Across these applications, image anomalies are broadly classified into two: semantic anomaly, a sample outside the 'normal' semantic distribution, and sensory anomaly, caused by unexpected pixel-level aberrations in an otherwise normal sample. These anomaly types are often handled with specialized approaches Jiang et al. (2022).

Prior work in deep AD is predominantly based on the familiarity hypothesis, where the anomaly is identified by the lack of *familiar* features in them Dietterich & Guyer (2022). Familiar features are the set of features the neural encoder has learned to represent meaningfully in the representation space. Inventive methods have been proposed to learn feature spaces where anomaly can be characterized by lack of familiar features, specifically representations of ViT fine-tuned on a related task have shown to excel in identifying samples out of train distribution Fort et al. (2021). The state-of-the-art AD method uses feature representation of a pretrained ViT backbone fine-tuned to classify normal samples from samples generated using a diffusion model prematurely early stopped while approximating the normal distribution Mirzaei et al. (2023).

Relying solely on the set of familiar features leads to two major issues in deep AD. Firstly, neural networks show 'unreasonable' generalize well beyond the training data Zhang et al. (2021), often showing invariance in representation to even OOD samples Jacobsen et al. (2018). This excessive invariance of representation well beyond the train distribution can lead to false negatives in familiarity based AD. Secondly, this paradigm does not account for anomalies caused by truly novel features not being represented meaningfully in the feature space through the learned encoding, also leading to false negatives.

Figure 1: Feature learning based AD methods succeed by detecting the presence and absence of *familiar* features in the test sample. Familiar features are the features the encoder learns to discriminate the normal samples from the background. The detection method fails for samples with *novel* features that the encoder is not trained to represent in the feature space.

While significant work has gone into countering the excessive generalization Mirzaei et al. (2023); Cohen & Avidan (2022); Tack et al. (2020) the latter issue needs a different modelling and strategy. In particular, current successful pre-trained embeddings might capture novel features poorly or not at all. Moreover, solving the excessive generalization often involves making assumptions on the nature of anomalies Hendrycks et al. (2018) or generating complex distribution Mirzaei et al. (2023) as background samples to control generalization.

We propose a novel approach for AD that addresses both these key issues by jointly modeling the lack of familiarity and presence of novelty towards anomaly detection. We use the features extracted by the encoder to compute familiarity and capture novel features as unexplained observations in the input space. A faithful explanation of the encoding enables inspection of features that were not meaningfully interpreted by the encoder. In this work, we use B-cos networks Böhle et al. (2022) to summarize the encoder into a faithful explanation of the encoding. We show that accounting for novel features for AD reduces the reliance on a complex background model to control generalization. While the two scores are not mutually exclusive, our experiments show that the latter adds to AD performance.

We evaluate the method across multiple benchmarks, handling both sensory and semantic anomaly types. We establish new state-of-the-art in most of the evaluated benchmarks. In particular, we show that by taking account of novel features, we reduce false negative anomalies by up to 40% across challenging benchmarks. Our experiments show that joint modeling eliminates the need for expensive background models and dense matching to improve AD performance. For sensory anomaly, the explanation is traced to the input, giving visually inspectable explanations. Since early layers of the backbone pre-trained on large datasets are frozen while training for AD tasks, we compute novelty with respect to these features for detecting high-level semantic novelty. In short, we make the following contributions:

- We define the idea of familiar and novel features in a test input in the context of anomaly detection. We propose a joint model for AD that accounts for the lack of *familiarity* and the presence of *novelty* in an input sample.

- We use the lack of encoder explanations to capture the novel features in the test input. This enables the sensory level and semantic level anomaly detection by detecting novelty at different hierarchies of the neural network. It gives visually inspectable explanations for sensory anomalies.

- Our method reduces the reliance on features and, hence the background samples. We show this effect by comparing the performance using two methods to generate background class (a) sampling from a normal approximation of normal train samples and (b) using a diffusion model.

## 2 RELATED WORK

This section discusses the methods developed to improve deep AD in the prior art. Most deep AD methods first train an encoder to learn a representation for normal samples and then use this representation to compute the anomaly score for a test sample Hojjati et al. (2022); Han et al. (2022). Hence, we categorize the popular methods as improving feature representation and test time detection. The survey shows that anomaly detection (though often synonymous with novelty detection) has often been solved by detecting the absence of familiarity.

**Improving feature learning for AD:** Perera *et al.* notes that using NN's in AD aims to learn robust feature space to define normalcy Perera & Patel (2019). Bergman *et al.* shows that using backbone pre-trained on large datasets substantially improves AD performance Bergman et al. (2020). Fort *et al.* demonstrates that transformers (pre-trained on large datasets) can significantly improve OOD tasks across different data modalities Fort et al. (2021). Investigating the trend of finding better representation space for AD Reiss *et al.* provides theoretical and empirical evidence to show that AD cannot improve indefinitely by increasing the expressiveness of networks Reiss et al. (2023). In fact, they show that there is a trade-off between expressiveness of features and sensitivity to anomalies.

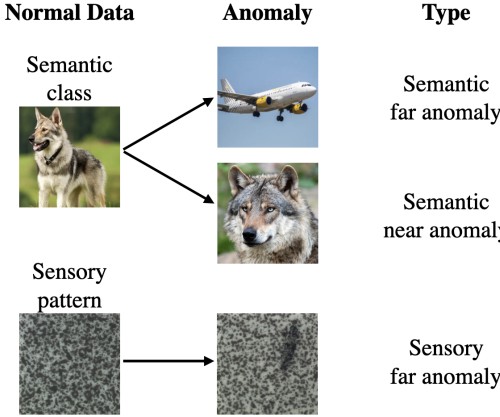

Figure 2: Illustration of predominant anomaly types considered in prior work.

AD performance has been shown to improve by controlling the generalization of the NN encoder using fine-tuning with, controlled outlier exposure, using real outliers, random images from the internet, or other samples from other datasets Hendrycks et al. (2019); Fort et al. (2021). Using GANs to generate outliers Kong & Ramanan (2021); Pourreza et al. (2021) shows further improvement over real images. Mirzaei *et al.* Mirzaei et al. (2023) use a prematurely stopped SDE model to generate background samples at the boundary of the distribution. A desirable property of AD is a reduced reliance on background class and minimal assumptions on the nature of anomaly. Our method shows robust performance with simple background approximation.

**Test time detection methods:** Popular detection methods for OOD detection and AD are: maximum softmax probabilities Hendrycks & Gimpel (2016), Local Outlier Factor Lin & Xu (2019), Gaussian Discriminant Analysis Xu et al. (2020) or nearest neighbor Bergman et al. (2020) to compute the similarity between the representations of normal samples and the test sample. Reiss *et al.* observe that these methods are opaque and non-interpretable. As anomalies are ambiguous, it is necessary to give explicit reasoning behind the criteria for detection. Our method explains the anomaly score when the anomaly is spatially local.

**Anomaly detection methods for sensory anomalies:** According to the different distribution shifts that cause them, anomalies are divided into sensory and sensory anomalies and semantic AD (Figure 2) Yang et al. (2022) Jiang et al. (2022). As sensory anomalies contain dense familiar features, its challenging to tackle via familiarity hypothesis Bergmann et al. (2019). Most methods use locally sensitive dense feature extractors such that a novelty in input can come only at the cost of lost familiarity Roth et al. (2021); Cohen & Hoshen (2020). With the increasing number of normal samples, the memory bank becomes exceedingly large, with it both the inference time and memory required. Roth *et al.* uses a coreset sub-sampling to reduce this effect Roth et al. (2021). Jiang *et al.* surveys methods for visual sensory anomalies and notes that most methods tuned for sensory anomalies perform poorly in detecting semantic outliers Jiang et al. (2022). Accounting for novelty in input could be key in bridging this gap in performance.

**The familiarity hypothesis:** For a regular NN, activation in the last layer for a novel class sample is usually much smaller than samples from training data. Vaze *et al.* suggests that this difference can be a good open-set-recognition Vaze et al. (2022). Neural encoders give dense representations when familiar features are present in the input and fail to give an equally dense representation for samples with novel features Tack et al. (2020). Dietterich *et al.* formulates this as the familiarity hypothesis and argues that familiarity-based detection is an inevitable consequence of representation learning in AD Dietterich & Guyer (2022). Previous efforts to create a hybrid model for AD unifies the approaches of generative modelling for regular training data and discriminating with respect to negative training data Grcić et al. (2022). We propose augmenting the feature familiarity score with a score that accounts for novelty in the input for AD.

## 3 FAMILIAR AND NOVEL FEATURES FOR ANOMALY DETECTION

In this section, we formally introduce the concept of familiar features in the context of AD. We show how most existing state-of-the-art methods are predominantly reliant on familiar features. We then define novel features in AD and argue that leveraging novel features can open up new potential in AD.

Consider the domain of an AD task, set $S$. Unlike a discriminative task, by defining a generative mechanism for normalcy, the AD task divides the set $S$ into two mutually exclusive and exhaustive subsets. The first subset is the samples generated by the generative mechanism: normal samples $N$, the complementary set is the background $B$, $S = N \cup B$. Note that, background samples are not to be confused with background pixels within images. In the case of image AD, $B$ contains all pixel configurations that are not generated by normal mechanism, hence containing all semantic and sensory anomalies.

We use $\hat{\phantom{x}}$ operator to denote the set of all features that can be extracted from a sample/set. For instance, the set of all features that can be derived from the domain $S$ is $\hat{S}$. Also, the set of all features that a function $f$ can meaningfully encode is $\hat{f}$.

**Feature representation based AD:** Consider an ideal general model for AD, where for a hypothesis $F$, the parameter $\theta^*$ learns a representation space over which a single layer discriminates all samples in $N$ from a portion of samples in $B$ used for training, say $b$. Let the features that the encoder can now detect from $S$ be the set $\hat{F}(\theta^*)$. The hypothesis class $F$ learns a representation where a linear classifier shatters the space $F(\theta^*, N) \cup F(\theta^*, b)$. It is important to note that even under these strong assumptions, there is no guarantee that $\hat{F}(\theta^*)$ can give representations that can discriminate samples of $N$ from the set $[B - b]$. Even with an oracle training (all features that discriminate samples of $N$ and $b$ are in $\hat{F}(\theta^*)$), the elements in $[B \hat{-} b]$ that are not in $\hat{F}(\theta^*)$ are not captured in the representation. The potential error in the representation of elements in $[B \hat{-} b] - \hat{F}(\theta^*)$ explains a significant failure mode of AD.

Furthermore, it is tempting to believe that increasing the size of the hypothesis class and feature representation layer increases the size of $\hat{F}$ for AD using the neural encoder. Theoretical and empirical observation in Reiss et al. (2023) reveals that an increase in $\hat{F}$ affects the sensitivity of the familiarity based AD methods. This shows the importance of going beyond improving $\hat{F}$ for AD.

**Familiar features:** In realistic scenarios, a network $F(\theta)$ is learned to discriminate a subset of $N$ say $n$, with $b$. Elements of $\hat{F}(\theta)$ are the familiar features. For a test input $x_{\text{test}}$ in $[B - b]$, the part of familiar features used to derive anomaly score in current AD methods ($F_{AD}$) is given by:

$$F_{AD} = \hat{F}(\theta, x_{\text{test}}) \cup \hat{F}(\theta, n_i) \forall n_i \in n \tag{1}$$

$F_{AD}$, are the set of all features used by current familiarity based methods. **$F_{AD}$ is a subset of familiar features $\hat{F}(\theta)$.**

**Novel features:** We define the novel features in input as the set of features in a test input that are not in $\hat{F}(\theta)$. We define a function $G$ to capture all novel features in a test sample. The features in the input $x_{\text{test}}$ that are not familiar to the encoder is a subset of $\hat{x}_{\text{test}}$. Also, the novel features captured by $G$, i.e. $\hat{G}$ are not in the familiar feature set: $\hat{G} \nsubseteq \hat{F}(\theta)$. Hence the novel feature set given an input $x_{\text{test}}$ and a trained encoder $F(\theta)$ is given by,

$$\hat{G}(x_{\text{test}}, F(\theta)) = (\hat{x}_{\text{test}} - \hat{F}(\theta)) \tag{2}$$

In summary, the $\hat{x}_{\text{test}}$ can contain features outside $F_{AD}$ that might be causing the anomaly. Capturing these features can help reduce false negatives. We present a method to capture features in $\hat{G}(x_{\text{test}}, F(\theta))$ using explanations and propose a joint model for AD using both familiar and novel features. We define normal and familiar features within the AD context, deferring a formal analysis of this phenomenon for future investigation.

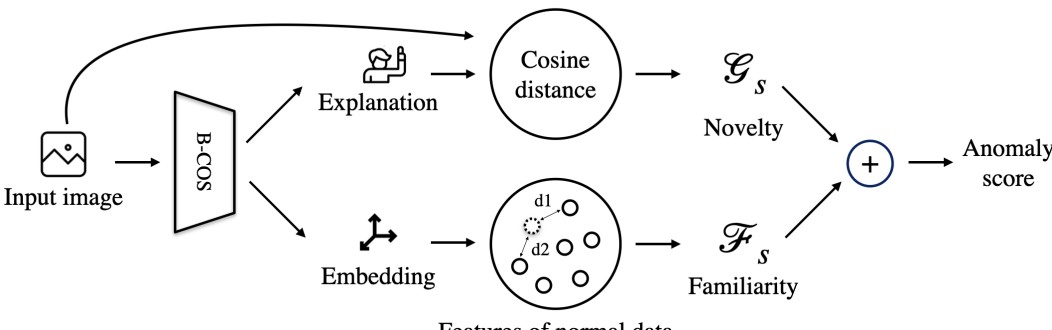

Figure 3: Figure shows the proposed pipeline. The top portion computes Explanation-based Novelty Score (ENS) and the bottom branch computes Familiar Feature based anomaly Score (FFS). The final score, novelty accounted anomaly score is a combination of both.

## 4 JOINT MODEL FOR FAMILIARITY AND NOVELTY BASED ANOMALY DETECTION

In this section, we present our method to capture novel features and account for them to compute anomaly scores for a test sample. We describe the proposed method to jointly model the lack of familiarity and presence of novelty for anomaly detection. The pipeline of the proposed AD method is shown in Figure 3. The feature extractor and the binary classifier are trained together to discriminate the normal class from the background class. The figure shows how, given a test sample, the familiarity and novelty scores are computed using the train features and the encoder explanation respectively.

**Familiar Feature based anomaly Score (FFS)**   In order to detect an anomaly by the lack of familiar features in a test sample, we use a mechanism similar to prior art. We first train an encoder $F$ to discriminate a subset of normal data $n$ from a subset of background data $b$ to get parameter $\theta$. Using $F(\theta)$ as encoder, we compute the distance between features of the test sample ($\hat{F}(x_{\text{test}})$) and the train normal samples ($\hat{F}(\theta, n_i) \forall n_i \in n$). We call this measure a Familiar Feature based anomaly Score (FFS). We compute $F(\theta, n_i) \forall n_i \in n$ and store them as the rows of matrix $M$. The FFS ($\mathcal{F}_s$) for input $x_{\text{test}}$ is computed as the sum of distances of the test feature to the two nearest train features ( following Mirzaei et al. (2023), details in Appendix 3). FFS score increases with the lack of familiar features in the test sample.

$$\mathcal{F}_s(x_{\text{test}}) = \|F(\theta, x_{\text{test}}) - M_0\| + \|F(\theta, x_{\text{test}}) - M_1\| \tag{3}$$

where $M_0$ and $M_1$ corresponds to the rows in $M$ closest to the test feature vector. This method requires the computation and storing of all train normal sample representations.

**Explanation Based Novelty Score (ENS)**   To capture the novel feature in a test input $\hat{G}(x_{\text{test}}, F(\theta))$ we use the explanation of a B-cos network. An encoder built with B-cos operator generates a reliable explanation of its computation. B-cos networks are neural networks where the linear layers are replaced by B-cos layers. For more details on the formulation and training of the networks, we refer the reader to Böhle et al. (2022). Operation of a B-cos layer at a node for an input $x$ and parameters $w$ leading to the node is given by

$$\text{B-cos}(x; w) = \|x\| \cdot \|w\| \cdot \cos(\angle(x, w))^B \cdot \text{sign}(\cos(\angle(x, w))) \tag{4}$$

Where $B$ is a hyper-parameter that influences the extent to which alignment between $x$ and $w$ contributes to the magnitude of the output. Using the B-cos transforms instead of linear transform removes the need for other explicit non-linearity while training the network. Hence, the only non-linearity in the network is dependent on the input. Given an input, B-cos network collapses into

a single linear transform that faithfully summarises the entire model computations. Moreover, the B-cos transform introduces alignment pressure on the weights during optimization. For the output of a node to be high, it requires that the input is aligned to the incoming parameters of the node ($\cos(\angle(x, w))$ is high). When the output of the network is high, the summarized linear layer is highly aligned to the input. Hence, using a B-cos network help generate a faithful explanation of the decision aligned to the parts of the image that contribute to activating the network's output.

We use the lack of this explanation as evidence of the presence of novel features. Equation 2 defines the novel features as the features in the test sample ($\hat{x}_{\text{test}}$) that are not familiar feature set ($\hat{F}(\theta)$). We quantify the size of this set using the cosine distance between $x_{\text{test}}$ and the explanation of the encoder explanation and use this to compute novelty score. B-cos encoder explanation gives a reliable summarization of network computation.

We approximate novel features ($\hat{G}(x)$) as that portion of the input features that the pre-trained B-cos encoder cannot align to. It corresponds to the feature in the input that the encoder fails to explain in the context of the decision. For a given input $x$ Böhle *et al.* denotes the explanation of an $L$ layer neural network as $\theta_{1 \to L}(x)$. We compute the Explanation based Novelty Score (ENS) denoted by $\mathcal{G}_s$ for a test input $x_{\text{test}}$, for an encoder with parameters $\theta$ as

$$\mathcal{G}_s(x_{\text{test}}) = \cos(\angle(\theta_{1 \to L}(x_{\text{test}}), x_{\text{test}})) \tag{5}$$

Note that the $\mathcal{F}_s$ score is computed using features alone while $\mathcal{G}_s$ does not rely on the encoded normal features. Finally, we compute the joint anomaly score as the sum of normalized familiarity and novelty score: Anomaly score for a given test sample $x_{\text{test}} = \mathcal{G}_s(x_{\text{test}}) + \mathcal{F}_s(x_{\text{test}})$.

## 4.1 ADAPTING TO ANOMALY TYPES

Since we use a backbone pre-trained on large data, the initial layers derive a wide range of features. With frozen initial layers, it becomes meaningful to check for novel features higher up in the neural layer hierarchy with respect to these features. For a test sample, Böhle *et al.* uses $\theta_{1 \to L}(x_{\text{test}})$ to visualize the explanation of the decision. The layers are collapsed from the input to the output node of a classifier of $L$ layers. We modify this formulation to capture the the portion of features that are explained given the final encoding, instead of computing the portion of input that explains the decision. That is, we compute $W_{l \to L}$ where $l$ is the layer at which we evaluate the novelty and $L$ is the final layer. Novelty of feature $F_i$, output by layer $i$ is computed as,

$$\mathcal{G}_s(f_i) = 1 - \cos(\angle(\theta_{(i+1) \to L}(f_i), f_i)) \tag{6}$$

For sensory anomalies, the value of $i$ is one, and Equation 6 becomes similar to the formulation in the prior art. Here, the choice of layer is a hyper-parameter to adapt to anomalies at different semantic levels.

## 5 EXPERIMENTS AND RESULTS

This section discusses the four experiments to evaluate the effect of accounting for novel features for computing anomaly scores. The first experiment evaluates the performance of the proposed novelty capture method in case of sensory anomalies. In the second experiment, we demonstrate the efficacy of the proposed framework on different AD benchmarks. We benchmark our method on eight different datasets to evaluate its overall effectiveness. The benchmark shows the efficacy of the method across different anomaly types, the sensory anomaly, and visually near and far semantic anomaly. In the third experiment, we show the effectiveness of the joint model in reducing false negatives. In the final experiment, we showcase how our method helps reduce reliance on background class.

## 5.1 EXPLAINING AT INPUT LEVEL FOR SENSORY ANOMALIES

In the first experiment, we evaluate the efficacy of using the explanation of the input for sensory AD. For this experiment, we use the MVtec dataset Bergmann et al. (2019), which has anomaly at the pixel level. The prior art that considers only feature familiarity without dense matching has reported

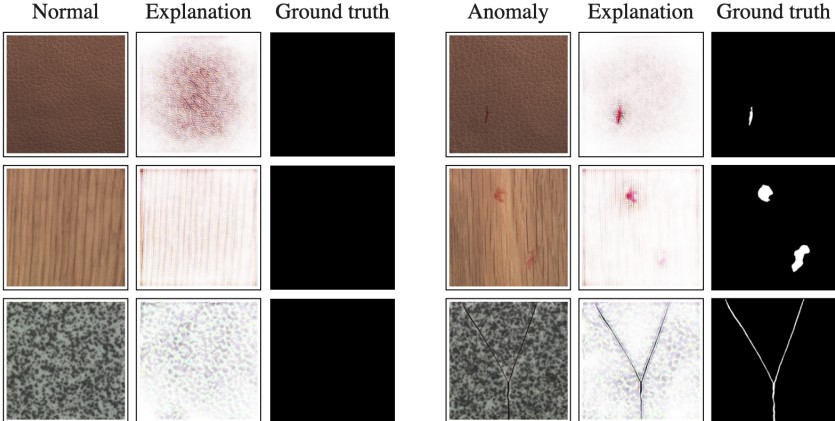

Figure 4: Test samples from MVTecAD dataset, and the explanation for being classified as normal using a B-cos model.

| Method | Datasets | | | | | | | | |
|--------|----------|--|--|--|--|--|--|--|--|
| | Semantic near AD | | | | Semantic far AD | | | Sensory AD | |
| | CIFAR-10 | CIFAR-100 | Flowers | Birds | FGVC | Cars | C10-100 | MVTec | Average |
| CSI Tack et al. (2020) | 94.3 | 89.6 | 60.8 | 52.4 | 64.6 | 66.5 | 76.1 | 63.6 | 71.0 |
| MSAD(ViT) Reiss & Hoshen (2023) | 94.1 | 93.0 | 98.6 | 93.3 | 81.3 | 85.7 | 79.5 | 85.5 | 88.8 |
| Transformaly Cohen & Avidan (2022) | 98.3 | 97.3 | 99.9 | 97.8 | 84.0 | 86.7 | 82.5 | 87.9 | 91.8 |
| PANDA Reiss et al. (2021) | 96.2 | 94.1 | 94.1 | 95.3 | 77.7 | 87.6 | 76.8 | 86.5 | 88.5 |
| PatchCore Roth et al. (2021) | 67.2 | 64.1 | 74.8 | 58.1 | 67.8 | 78.3 | 67.2 | 99.1 | 72.1 |
| FITYMI Mirzaei et al. (2023) | 99.1 | 98.1 | 99.9 | 98.5 | 88.7 | 90.8 | 89.4 | 86.4 | 93.8 |
| Our method | 99.3 | 98.5 | 99.9 | 98.7 | 89.3 | 90.5 | 91.1 | 89.3 | 95.3 |

Table 1: The performance of the proposed method for semantic anomaly detection methods (AU-ROC) in the AD setting on different datasets. The best performance of the best-performing model is bold, and the second-best method is underlined.

relatively lower performance in this task compared to its performance on other benchmarks. The MVtec dataset has pixel-precise annotations of all anomalies to compare the explanations.

For every normal class in the dataset, we fine-tune a B-cos ViT backbone pre-trained on the ImageNet-1K dataset, with a two-class classification head (one for normal and the other for background). Unless mentioned otherwise, the value of B used in B-cos network across all experiments is 1.5. We use data samples from the normal approximation of the normal class as the background. Images of the class follow a complex distribution, and a normal distribution cannot capture this complexity. The classifier is trained to discriminate this error in approximation. More details of this choice are discussed in the section 5.3. We use standard training procedures as in Mirzaei et al. (2023) without any augmentation for fine-tuning the classifier.

For a test input $x_{\text{test}}$, explanation for the classification is computed as described in Section 4: $(\theta_{1 \to L}(x_{\text{test}}))$. Figure 4 shows the explanation in the input generated by the B-cos model for the anomaly branch of the classification head. This is the explanation computed for the decision that the input sample is an anomaly. This shows how the method can not only detect but also explain the anomaly. We compute the ENS score on MVTec and report the same as the anomaly score for comparison with other familiarity based methods (Figure 7(a)). This improved performance comes without using feature representations of normal samples. That is, the performance improvement comes with a reduced memory (the memory to store train normal features) and computation cost (of computing the K nearest neighbor).

## 5.2 BENCHMARKING ACROSS DIFFERENT AD TASKS

For benchmarking on different types of anomalies, like in the previous experiment, for every normal class of every dataset, we fine-tune a B-cos ViT backbone pre-trained on the ImageNet-1K dataset with a two-class classification head. We use data samples from the normal approximation of the normal class as the anomaly. The proposed method gives a hyper-parameter to control the semantic level at which the anomaly is computed (variable $i$ in Section 4.1). We do not tune this for each

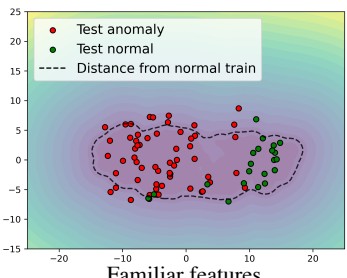 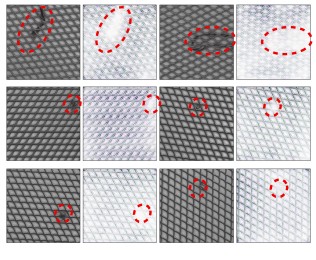 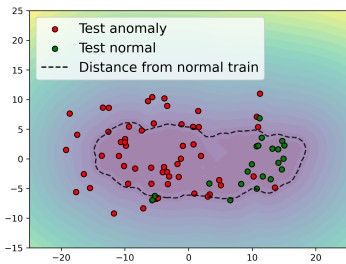

Familiar features     Novelty as lack of explanation     Novelty accounted familiar features

Figure 5: The PCA plot shows the normal test and anomaly test samples plotted on the two principal components. The first plot is the PCA using familiar features and the second is with novelty score added to the features. The contour shows the sum of distances to the two nearest train normal samples. The samples shown are the ones that give maximum deviation from train normal on adding novelty score.

dataset to optimize the performance. For a fair comparison across benchmarks, we use $i = 0$ for the sensory anomaly (the anomaly is in pixel level), $i = 6$ for all far anomaly (the anomaly is at low-level visual features), and $i = L - 1$ for semantic near anomaly (the anomaly is at high-level semantics). The further exploration of this parameter is left for future work.

Table 1 shows the performance of various prior-art benchmarked against the proposed method. The performance of some of the datasets like CIFAR-10 and Flowers are saturated (above 99%). On average, our model outperforms the next best model by more than 2.5%.

On the more challenging benchmark of near semantic AD (CIFAR-10 vs CIFAR-100 Mirzaei et al. (2023), where the closest classes of CIFAR-100 corresponding to each class of CIFAR-10 is picked for test), our method outperforms the familiarity based method by 1.0% establishing the new SOTA (details in Appendix 2. Furthermore, the table shows an interesting trade-off of performance across the two challenging tasks of near semantic AD and sensory anomaly the best-performing methods are different. While PathCore outperforms FITYMI by a margin of more than 13% on MVTec, FITYMI outperforms PathCore by more than 22% on near-semantic AD. This shows how one method is tuned for semantic anomalies and the other for sensory anomalies. Using novelty on the familiarity gives a more consistent performance across the two tasks. The gap in performance across the two tasks shows a scope for improvement in computing novelty.

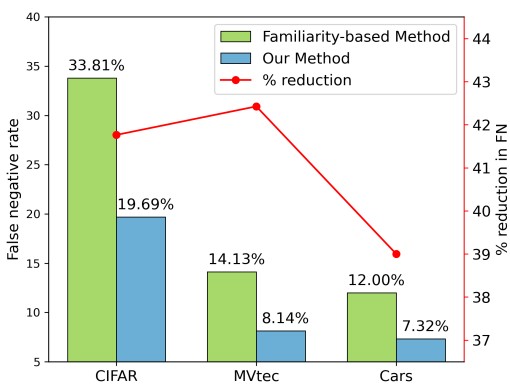

Figure 6: Comparing the false positives across different datasets with and without accounting for novelty. Y-axis: left shows the rate of FP, and the right side shows the % reduction in FP.

**Reduced False negatives:** Methods that use only familiar features will have issues with anomalies caused by truly novel features and hence produce false negative predictions. In this experiment we ablate familiarity and novelty branch to evaluate the role of incorporating novelty into the scoring mechanism in the false positive rate on three benchmarks: Stanford-Cars (semantic far AD), Cifar10-100 (semantic near AD), and MVTec (sensory AD). Note that FFS scoring is simlar to Mirzaei et al. (2023) and compared with addition of ENS to compute novelty. Furthermore, False Negative rate is an important characterization of an AD method in high-risk applications. To compute the same, we convert the anomaly score into classification. We use an oracle to find the optimum threshold for each class of each dataset. The results show that (Figure 6) accounting for novelty in input reduces the number of false positives by about 40% across the different anomaly types. This validates the hypothesis in Dietterich & Guyer (2022) that AD by relying solely on familiar features can lead to missing the anomalies caused by novel features.

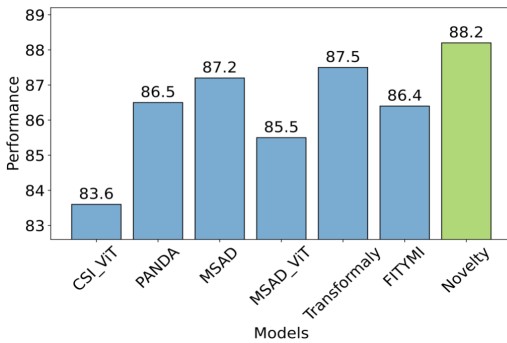 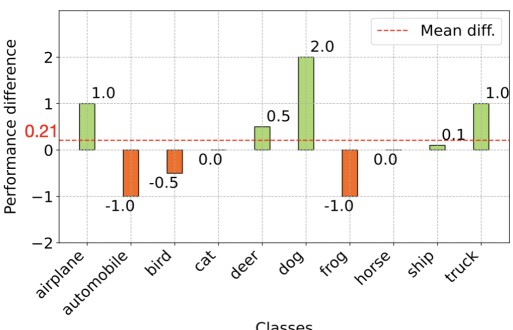

Figure 7: (a) Performance of AD models tuned for semantic AD on MVTec AD dataset compared against novelty based scoring (ENS).

(b) Difference in absolute AUROC using diffusion model and normal approximation for background class on near semantic AD.

Figure 5 shows further analysis of the reduction of false negatives in a challenging class of the MVTec AD dataset. It shows the features of normal test samples and test anomalies in their principal component space. The second plot is the features added with the sign-corrected novelty score. The color of the contour at each point shows the sum of the distances to the two nearest normal train samples. The samples and explanation are the ones that showed the highest difference by incorporation of novelty into the computation. The evidence of novel features is captured by Equation 5. The PCA plots show how accounting for novelty moves the anomaly samples further away from the normal train compared to the normal test samples. The figure shows how accounting for novelty as lack of explanation helps reduce such false negatives.

## 5.3 REDUCING THE ROLE OF BACKGROUND CLASS IN ANOMALY CLASSIFICATION

The impact of background class needs to be minimal for AD to be robust to the diverse set of anomalies encountered in the real world. Hence it is useful to have non-discriminative learning in AD. Mirzaei et al. (2023) uses a score-based generative model trained and prematurely early stopped on the normal samples. Appendix 1 shows evidence for the need to reduce reliance on the background class. Furthermore, recent work shows adding generated images improves ID performance on standard benchmarks Azizi et al. (2023). Generated images even show promising performance in replacing real images for training image classification tasks Sariyildiz et al. (2023). Stopping the generator training at the right point is vital to ensure the background samples are not part of the distribution.

We use a simple normal approximation of the normal class. That is, we compute the mean and co-variance of the normal dataset and sample from that distribution to generate an anomaly. Figure 7 shows how the model trained with normal approximation matches and almost outperforms the model trained with a diffusion model-based generation of the background class. Using normal approximation actually gives a mean improvement of about 0.2%, making it the preferred alternative. This reduced complexity can also come in handy when deploying AD in practice.

## 6 CONCLUSION

This paper highlights the need for Anomaly Detection (AD) to go beyond *familiar* features and incorporate *novel* features into the model. We are inspired by the 'familiarity hypothesis': AD methods that rely solely on familiar features cause consistent false negatives when anomalies are caused by truly novel features that are not well captured by the pre-trained encoding. Hence, we have proposed a method to capture truly novel features as unexplained observations and show that accounting for them reduces false negatives in AD. The proposed method establishes state-of-the-art results on multiple benchmarks across different anomaly types. The method also reduces the reliance on background class, allowing the use of simpler approximation in future work. We believe further research to capture novel features in test input will continue to improve anomaly detection and related tasks like Novel Class Discovery, Out-of-Class detection, and Out-of-Distribution Detection.

## REPRODUCIBILITY STATEMENT

We will release the code for our experiment and code for background data generation, and all trained models. The experiment details for familiar feature based AD are similar to the state-of-the-art method Mirzaei et al. (2023). The B-cos model implementations are taken from the official repository of Böhle et al. (2022).

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
