# Supplimentary material for submission titled: Going Beyond Familiar Features for Deep Anomaly Detection

## 1 Reliance on background class

Figure 1 shows the training profile of anomaly detection on the challenging formulation of Cifar-10. For each class of Cifar-10 the most challenging class from Cifar-100 is selected as test anomaly. The training is done following the method in Mirzaei et al. (2023), such that for each normal class, we train a ViT backbone to discriminate the normal sample from samples generated with an diffusion model trained to approximate the normal distribution.

The first two plots show the train and test loss for each of the ten classes in Cifar-10. The train and test loss shows the successful reduction of empirical risk in the classification task for which the encoder is getting trained. The third plot shows the AD performance on the held out test anomaly. We can see that for some anomaly classes like 'lizard'. the AD performance improves with the reducing train and test losses.

Considering the anomaly classes shown separately in the fourth plot, we can clearly see how for some anomalies, like class 'pick-up truck', the AD performance reduces with improved training. Despite the reducing loss at the classification task, the AD performance reduces. This shows the poor alignment between the representation learning task and the AD task. Hence designing background classes for any normal sample, without making assumptions on the nature of anomaly is a challenging task. Reducing the reliance on background class for feature learning is a desirable property for AD methods.

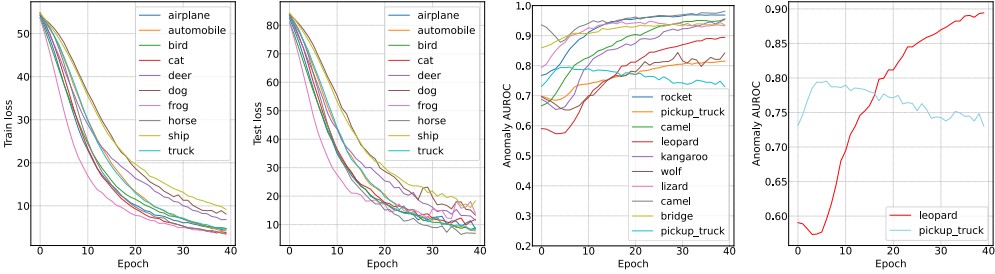

Figure 1: Training profiles of the ten AD backbones

## 2 Cifar-10 vs 100 near ND semantic anomaly Evaluation

In earlier AD benchmark, where the classes from different dataset are considered anomalies, a Cifar-10 vs Cifar100 benchmark follow that: category within CIFAR-10 is identified as the standard or normal class, while all other categories are classified as far anomalies. Mirzaei et al. (2023) presents a near AD evaluation, where the class from CIFAR-100 closest to normal class in CIFAR-10is used for evaluating detection process. It's the class associated with the lowest test AUROC in anomaly detection.

As mentioned in the 3rd para of section 5.2, we use this framework to evaluate the near ND benchmark. The closest classes of CIFAR-100 is the anomaly corresponding to each class of CIFAR-10.

To evaluate a model on this benchmark, first, the model is trained on each of the classes of Cifar-10 as a normal class. The model trained on a class, say class 'cat' from CIFAR-10, uses all samples from the train split of class 'cat' as normal for training. The model is then evaluated on all classes of CIFAR-100 one after the other, each as an anomaly with the test samples of class 'cat' as test normal samples. Then, the most challenging class (least AUROC) from CIFAR-100 is selected as the test anomaly. This is repeated for all 10 classes of CIFAR-10. Correspondingly, we get a near anomaly from CIFAR-100. Below are the normal class near anomaly pairs following the format 'CIFAR-10 normal class':CIFAR-100 anomaly class:

C10-100 = {'airplane': 'rocket', 'automobile': 'pickup-truck', 'bird': 'kangaroo', 'cat': 'rabbit', 'deer': 'kangaroo', 'dog': 'wolf', 'frog': 'lizard', 'horse': 'camel', 'ship': 'bridge', 'truck': 'pickup-truck'}

The intuition here is that any other class in the Cifar-100 as anomaly would yield a better AUROC score, making this the most challenging benchmark.

## 3 CHOICE OF K IN K-NEAREST NEIGHBOURS FOR FFS

We choose K=2 for the K-nearest neighbour model for familiar feature based anomaly detection. This design choice is taken from prior art for a fair comparison Mirzaei et al. (2023). Furthermore, the method is robust to the change of the nearest neighbor to a large extent as shown by Mirzaei *et al.* .

| Experiment | Dataset | C | | | | |
|:---:|:---:|:---:|:---:|:---:|:---:|:---:|
| | | k=1 | k=2 | k=3 | k=4 | k=5 |
| 1 | CIFAR-10 | 99.0 | 99.1 | 98.9 | 98.9 | 98.7 |
| 2 | CIFAR-10v100 | 89.8 | 90.0 | 90.0 | 90.0 | 89.7 |

## REFERENCES

Hossein Mirzaei, Mohammadreza Salehi, Sajjad Shahabi, Efstratios Gavves, Cees G. M. Snoek, Mohammad Sabokrou, and Mohammad Hossein Rohban. Fake it until you make it : Towards accurate near-distribution novelty detection. In *ICLR*, 2023. 1, 2