# OpenReview forum: "Going beyond familiar features for deep anomaly detection"
_ICLR.cc/2024/Conference — Submitted to ICLR 2024_

### Official Review · Reviewer_Qn1L · 2023-10-23

**Soundness:** 3 good
**Presentation:** 2 fair
**Contribution:** 2 fair
**Rating:** 5
**Confidence:** 4

**Summary:**

The manuscript deals with anomaly detection. Previous works model anomaly detection according to the familiarity hypothesis in which the absence of training features indicates anomalous input. The manuscript extends this line of work by complementing familiarity-based AD with novelty-based AD. Features are novel if they are unexplainable according to the observed data. In practice, explainability is captured by previously introduced B-cos networks, while familiarity is modelled using a memory bank of training features.
The resulting hybrid formulation yields fewer false-positive responses in different relevant experimental setups.

The manuscript claims the following contributions:

C1.  Formal definition of familiar and novel features in the context of anomaly detection.

C2. A hybrid model for AD that accounts for the lack of familiarity and
the presence of novelty in an input sample.

C3. Strong experimental results with reduced incidence of false positive responses.

**Strengths:**

S1. The manuscript deals with an important issue of anomaly detection.

S2. Defining novel features according to the absence of explainability is an interesting idea.

S3. The presented experimental results are good.

**Weaknesses:**

W1. The manuscript claims a formal definition of familiar and novel features as one of the main contributions. However, the presentation in Sec.3 is rather poor - definitions of familiar/unfamiliar features are bundled together, the notations are non-standard, and the whole section is hard to comprehend. Claimed definitions of familiar and novel features should be neatly formulated with appropriate definitions, lemmas, and theorems as in [a].

W2. The proposed method has design choices that are not well explained. E.g. why are two nearest train features used in Eq. 3? What are the benefits/issues of using more or less neighbours?

W3. The definition of the final anomaly score should be clearly stated together with the corresponding equation.

W4. The contribution of each component in the proposed hybrid anomaly score should be ablated.

W5. Missing relevant related work which introduces hybrid formulation for anomaly detection [b].

[a] Zhen Fang, Yixuan Li, Jie Lu, Jiahua Dong, Bo Han, Feng Liu: Is Out-of-Distribution Detection Learnable? NeurIPS 2022.

[b] Matej Grcic, Petra Bevandic, Sinisa Segvic: DenseHybrid: Hybrid Anomaly Detection for Dense Open-Set Recognition. ECCV 2022.

**Questions:**

See weaknesses.

---

> ### Author Response · Authors · 2023-11-17
> **Thank you for your positive feedback and comments.**
>
> Thank you for the constructive feedback and comments. We have answered the questions and edited the draft to incorporate the clarifications in the writing.
>
> W1. We recognize the importance of formalizing the definition of these concepts. This can not only help AD but also tasks like Open set recognition, anomaly detection and novel class detection. We revised Section 3 to highlight the definitions from other discussions. Our primary aim is to define the idea of familiar and novel features in a test input in the context of anomaly detection and illustrate the practical implications. Taking forward the hypothesis in [2], a future work that approaches the theoretical foundation can help the field.
>
> W2. This design choice is taken from prior art for a fair comparison. We keep design choices of familiar feature branch same as the prior art [1]. Furthermore, the familiarity branch is shown to be robust to change of the nearest neighbor to a large extent. Below are the results of [1] to value of k. We add this to the appendix.
>
> ***
>           Dataset                   KNN
>                     k=1     k=2     k=5     k=10     k=50
> ***
>          CIFAR-10   99.0    99.1    98.9    98.9     98.7
>          C10v100    89.8    90.0    90.0    90.0     89.7
>
> ***
>
>
> W3. Thank you for the suggestion, have added that to the section. The anomaly score computed as, novelty and familiarity score combined is highlighted.
>
> W4. Figure 6 is an ablation of the familiarity and novelty branch. Also the comparison of [1] to our method also shows the effectiveness of incorporating novelty. Except for the choice of encoder. As mentioned in section 4 “In order to detect an anomaly by the lack of familiar features in a test sample, we use a mechanism similar to prior art.” The improvement in performance of the evaluated framework from [1] can be attributed to accounting for novelty.
>
> W5. Thank you for the reference. We have added it to the related work as other notions of hybrid anomaly detection. [b] proposes a method to unify the approaches of generative modelling for regular training data and discriminating with respect to negative training data for anomaly detection. This is an interesting idea yet different from the notion of familiarity hypothesis explored in our work.
>
> [1] Fake it until you make it : towards accurate near-distribution novelty detection
> [2] The Familiarity Hypothesis: Explaining the Behavior of Deep Open Set Methods

---

> > ### Comment · Reviewer_Qn1L · 2023-11-22
> > **Post author response**
> >
> > I thank the reviewers for the response. However, some of my concerns are still not addressed. For example, the answer to (W4) references Fig.6 which only compares familiarity-based AD with the hybrid formulation. What about novelty-based AD? Does it even contribute? This is a major concern since the manuscript still doesn't show that hybrid formulation performs better than both familiarity-based AD and novelty-based AD. Additionally, the manuscript is still hard to read. I will wait for the post-rebuttal comments of other reviewers, but at this point, I'm leaning towards decreasing my score.

---

### Official Review · Reviewer_oe8U · 2023-10-24

**Soundness:** 3 good
**Presentation:** 2 fair
**Contribution:** 3 good
**Rating:** 3
**Confidence:** 3

**Summary:**

This paper examines the distinction between novel and familiar features in the context of anomaly detection and proposes an AD methodology which jointly models both novel and familiar features in its scoring.

**Strengths:**

Considering performance for both semantic and sensory anomaly detection with a single methodology is beneficial and also often neglected in other works, and this methodology does appear to give the best overall performance across both types of AD tasks.

Anomaly detection via explanation is a relatively under-explored approach.

**Weaknesses:**

Section 3 can summarised in just one sentence that comes near the end: "In summary, the xˆtest can contain features outside FAD that might be causing the anomaly." As such, it is hard to justify the very heavy use of notation, the explanations of novel and familiar features are longwinded and their definitions are also not very clear. This section could be made more clear and concise, defining exactly what differentiates familiar and novel features.

Some aspects of the experiments are not particularly clear (see questions 1 and 2).

In Figure 5, by choosing to present only the samples that give the maximum deviation from the training set upon adding the novelty score,  the figure is artificially amplifying the effect of the novelty score. It would be better to randomly sample points to give a more unbiased picture of the effect of the novelty score.

Section 5.2 is not well argued as it is not clear that this methodology has reduced the role of the background class, as there are no experiments that measure performance with and without training with the background class. Instead, this paper simply changes the way the background class is generated.


Edit: In consideration of other reviews, I update my decision to reject. The authors may improve the rigor of the explanation and intuition behind familiar and novel features and the how this leads to their methodology to improve the paper.

**Questions:**

1. What were the classes chosen for the near-AD experiments? How were these classes chosen? Were other class splits tested and, if so, how did performance vary between them?

2. What is in the intuition of using the normal approximation to simulate anomalies. How about other augmentation strategies used in anomaly detection in previous works?

3. What is the intuition that ensures that using explanations from the B-cos network focuses on identifying novel features in its scoring. How are we sure it is not relying on the deviation in familiar features between normal and anomalous samples, just like the familiarity score module?

---

> ### Author Response · Authors · 2023-11-17
> **Thank you for your positive feedback and comments.**
>
> Thank you for recognizing the novelty and contributions in the paper and the constructive feedback. We have edited the draft to incorporate the clarifications in the writing.
>
> **Question 1**
> Thanks for seeking clarification on the experimental details. We have added the details in the appendix. This evaluation strategy and choice of classes is the same as the prior art [12]. As mentioned in the 3rd para of section 5.2, for the near ND benchmark, we choose CIFAR-10 vs CIFAR-100, where the closest classes of CIFAR-100 is the anomaly corresponding to each class of CIFAR-10.
> To evaluate a model on this benchmark, first, the model is trained on each of the classes of Cifar10 as a normal class. The model trained on a class, say class 'cat' from Cifar-10, uses all samples from the train split of class 'cat' as normal for training. The model is then evaluated on all classes of Cifar-100 one after the other, each as an anomaly with the test samples of class 'cat' as test normal samples. Then, the most challenging class (least AUROC) from Cifar-100 is selected as the test anomaly. This is repeated for all 10 classes of Cifar-10. Correspondingly, we get a near anomaly from cifar100. Below are the normal class near anomaly pairs following the format cifar10_normal_class:{c100_anomaly}:
>
>        C10_100 = {'airplane': 'rocket', 'automobile': 'pickup_truck', 'bird': 'kangaroo', 'cat': 'rabbit', 'deer': 'kangaroo', 'dog': 'wolf', 'frog': 'lizard', 'horse': 'camel', 'ship': 'bridge', 'truck': 'pickup_truck'}
>
> The intuition here is that any other class in the Cifar-100 as anomaly would yield a better AUROC score, making this the most challenging benchmark. For instance, [12] reports a 20% drop in performance of previous SOTA, PANDAS[11] on this benchmark compared to global Cifar-10-100 evaluation.
>
> **Question 2**
> The key motivation here is to compare the method's performance between the SOTA background class (diffusion model to approximate normal semantic distribution) and simpler approximation (the distribution statistics are the same as the normal class with no guarantees on the semantics). There is a correlation between AD performance and the FID distance between the normal and the background used to train the model.   The two are negatively correlated until the FID is too low (normal~background). The intuition here is that the closer the background is to the normal (or more challenging the discriminative task for the encoder ), the better suited the learned feature is for AD. In the paper, we say that we hypothesize that reduced reliance on encoder derived features in our method can be the cause of this reduced reliance on the choice of background class.
>
>     CIFAR10 - 100 (Near ND trained with SDE)
> ***
>     FID          400             300               200            100              50
>     AUC         82.3             83.4              86.0           88.5            89.2
> Using a normal approximation as background is a simpler, and comes at much lesser computation (~zero training) giving an FID of : 523.7 across classes. For familiarity methods, using high quality approximations gives an improvement of more than 7% on the near-ND benchmark[12]. Proposed method shows the performance is consistent across the 2 evaluations showing the reduced reliance on background for AD by incorporating novelty.
>
> The table below shows the AD performance with other standard generators for generating background for FFS. We picked the best method from this for our experiment.
>
>                           Baseline-VAE    Baseline-Gan       BigGAN    StyleGAN-ADA  StyleGAN-XL  DenseFlow     SDE
>     CIFAR10                 93.8             94.5             96.9         97.8         96.5        96.4       99.1
>     MyTecAD                 68.8            71.8              76.4         80.5         73.2        78.93      86.4
>
> [11] PANDA: Adapting Pretrained Features for Anomaly Detection and Segmentation
> [12]  Fake it until you make it : towards accurate near-distribution novelty detection
> [13] Opengan: open-set recognition via open data generation
>
> **Question 3**
> Explanation of B-cos gives a faithful summary of the encoder computation. Lack of this explanation implies the feature has not been used in the encoding (represented in the feature space). Absence of explanation is captured as the cosine distance between input features and the B-cos encoder explanation.
> Please note that computing novelty scores does not involve the use of train normal features or train normal explanations. It is computed between the input sample and the corresponding explanation as the distance of the test sample from the explanation of the encoder.
> This is enabled by the explanation faithful to the computation of the encoder (Bcos)  and has a one-to-one correspondence with input features (enabling inspection for absence of features). In this framework lack of explanation means that the encoder has not captured that feature well.

---

### Official Review · Reviewer_vwNp · 2023-10-31

**Soundness:** 2 fair
**Presentation:** 1 poor
**Contribution:** 2 fair
**Rating:** 3
**Confidence:** 4

**Summary:**

This paper proposes hybrid framework with a familiarity branch and novelty branch for anomaly detection. The familiarity branch is based feature comparison, and the novelty branch is based on B-cos network’s explanations.

**Strengths:**

Introducing novel features for anomaly detection is interesting, it seems that it can be further studied.

**Weaknesses:**

1. I think the novelty branch and familiarity branch are similar. The familiarity branch detects anomalies by comparing test samples features with normal samples features, while the novelty branch relies on anomalies with different explanations with normal. I feel this is just transfer from the feature space to the B-cos network’s explanations.

2. Bad academic terminology: “the complementary set is the background B…”, it’s weird to call anomalies as background set. In the defect detection (or sensory anomalies in paper), background generally refers to normal regions. Even for semantic anomaly, it’s also weird to call anomalies as background. The description of familiar and novel features is also hard to follow.

3. Evaluation on sensory anomaly dataset is not enough, only MVTec. Moreover, your method performs poorly on sensory anomalies and cannot compared with the anomaly localization methods. So, just focus on the semantic anomaly, your method visual inspectable explanations are not very valuable.

**Questions:**

1. According to the assumption in the paper, untrained anomalies don’t belong to the familiar space, so these anomalies will be determined as false negatives? Moreover, anomaly appearances are usually different from normal, although the network don’t generate representations in the familiar space, the network will generate representations that are different from normal. So, anomalies can also be determined through familiarity branch.

2. Thus, I think you should demonstrate that kind of anomalies will be determined as false negatives by the familiarity branch, and the novelty branch can solve this. Although Figure 6 shows that the novelty branch can reduce false negative rate, this figure is likely unfair, as we don’t know what the familiarity-based method specifically refers to. It needs to your familiarity branch to be fair.

3. Figure 5 still cannot explain the effectiveness of your method. Adding novelty only increases the distance of anomalies that could be classified in the first figure, but the confusion regions in the first image still exists in the third figure.

---

> ### Author Response · Authors · 2023-11-17
> **Thank you for your review. We respond to the questions and points raised below:**
>
> Thank you for the feedback. We address a few points raised in the weakness and answer questions on details of the method. Updated draft to reflect clarifications.
>
> **Weaknesses**
>
> 1: We would like to clarify that novelty branch does not rely on the explanation of normal samples, and the proposed method does not involve computing explanations for normal samples. Novel features are defined as the input features the encoder does not meaningfully represent in the feature space, captured with the distance of the test encoder explanation to the test sample. Contrasting Equations (4) and (5) show that familiarity uses normal features while novelty computation at inference does not rely on normal samples, its features, or its explanations. (Section 3: novel features, Section 4 ENS computation). Just transfer from the feature space to the B-cos network’s explanations is a mischaracterization of the method.
>
> 2: Thank you for highlighting the potential confusion in the definition. We have included clarification in the definition to avoid confusing background samples with background pixels within images. What we call background is different from what previous methods have called ‘training outliers’ [2], ‘outliers’ [5] [1],  irregular samples[4], or just anomalies[1]. In prior art, background has been used to denote novel distributions/classes [2][3]. We use background for denoting complimentary set not in the normal distribution, including novel semantic classes, abnormal images, and random pixel configurations.
> Furthermore, Appendix 1 shows how training on a subset of background actually can reduce test AD performance in some cases (even on SOTA-generated background), motivating not just calling them anomalies.
>
> 3: Explainability is a desirable property in AD[6], and having a method that reliably explains anomalies can be a helpful addition. ML explainability has focused on reliability of the explanation methods[11], and the need to going beyond saliency explanation[12]. A limitation of feature-KNN methods is the lack of explanation [7]. Explanation research has application beyond localizing anomalies as well. Our evaluation of sensory anomaly is the same as the benchmarks used in [1] the SOTA familiar feature-based AD method. The paper states that we do not match the sensory anomaly SOTA with dense feature matching and shows a comparison with the familiar feature KNN methods. MVTec is one of the most popular and widely used industrial anomaly datasets suited for sensory anomaly detection [8][9]. Localization papers have also used it as the only sensory AD benchmark [8].
>
> **Answers**
> 1) The paper does not assume that ‘untrained anomalies’ will be false negatives. [10] Present Familiarity Hypothesis: Existing AD methods succeed by detecting the presence and absence of familiar features (features the encoder has learned to represent) rather than the presence of novelty. The paper shows how this is an essential shortcoming of feature based anomaly detection.
> Anomalies represented differently from normal samples in the feature space can be captured using the familiarity branch. The novelty branch helps capture the features the encoder is not trained to represent meaningfully in the feature space.  Our experiment shows how incorporating novel features into computing anomaly scores improves AD performance, especially reducing false negatives.
>
> 2) Evaluation on different types of benchmarks show the AD performance improvement on different  'kind of anomalies'. We compare AD performance using the familiarity branch alone and after incorporating novelty across 3 AD types to show the effect of incorporating novel features into AD computation.
> 'Likely unfair': We have now clarified the description in section 5.2. Please note that, the figure shows how incorporating novelty to the familiarity branch reduces FNR. We use a mechanism similar to SOTA to detect the lack of familiar features[1].
>
> 3) The figure shows two dominant axes of the PCA plot. It is shown to visualize the inner workings of incorporating novelty. The improvement in performance and the reduction in FN is the quantitative proof that incorporating novelty is effective in improving the anomaly score shown as distance in the plot.
>
> [1] Fake it until you make it : towards accurate near-distribution novelty detection
> [2] Opengan: open-set recognition via open data generation
> [3] The pascal visual object classes challenge: a retrospective
> [4] G2d: generate to detect anomaly
> [5] Deep anomaly detection with outlier exposure
> [6] A survey on explainable anomaly detection
> [7] Sub-Image Anomaly Detection with Deep Pyramid Correspondences
> [8] Towards Total Recall in Industrial Anomaly Detection
> [9] PaDiM: a Patch Distribution Modeling Framework for Anomaly Detection and Localization
> [10] The Familiarity Hypothesis: Explaining the Behavior of Deep Open Set Methods
> [11] Towards A Rigorous Science Interpretable Machine Learning
> [12] The (Un)reliability of saliency methods

---

### Comment · Area_Chair_mJm9 · 2023-11-22
**Discussion Period Ending - Please Engage**

Dear Reviewers,

The authors have responded to the reviews. Please read over the responses and take this opportunity to clarify any further issues before the end of the discussion period today (Nov 22 AOE). Do update your reviews accordingly as well to note you have read the response.

Thanks,
AC

---

### Meta-Review · Area_Chair_mJm9 · 2023-12-06

**Metareview:**

This paper introduces an anomaly detection algorithm based on the idea of novel features being the cause of anomalies, which is in addition to current methods that rely on what the paper terms as familiar features. These novel features are identified using previously proposed B-cos networks, and a score based on these novel features are add to a score on familiar features for anomaly detection. The proposed algorithm is evaluated on a range of anomaly detection settings where it outperforms the current state-of-the-art on average.

Reviewers agree that the paper proposes an interesting and new perspective of using novel features to detect anomalies, where these features can be derived using the absence of explainability. While experimental results are promising, an ablation study showing the advantage over purely novelty-based anomaly detection would help motivate combining the scores. More importantly, all reviewers had concerns about the presentation, in particular, the lack of a precise definition of familiar vs novel features which is core to the method; there are also other concerns about how the work is described in view of norms in the anomaly detection literature (e.g. use of 'background class' instead of 'anomalies'). The AC also thinks the overall methodology (including how the model is trained, and with what data - are labeled anomalies required?) and in particular the explanation of how the explanations from the B-cos network result in novel features is not clear, which may have resulted in multiple reviewers being confused on how it works; the authors attempted to clarify this in their responses but reviewers were not convinced. During the discussion, reviewers unanimously agreed that the paper was not ready for publication due to the presentation issues - especially lack of clarity on definitions and how the novel branch works. The AC agrees and recommends the paper be rejected this time; that said the research direction is interesting and the AC suggests that the paper be revised accordingly before resubmitting.

**Justification For Why Not Higher Score:**

- Lack of precise definitions on familiar and novel features
- Lack of clarity on how/why the novelty branch works
- Lack of ablation study to show superiority over novelty-based AD

**Justification For Why Not Lower Score:**

N/A

---

### Decision · Program_Chairs · 2024-01-16

Reject